# Impact of Nanolayered Material and Nanohybrid Modifications on Their Potential Antibacterial Activity

**DOI:** 10.3390/nano12162749

**Published:** 2022-08-11

**Authors:** Hasna Abdullah Alali, Osama Saber, Mahmoud Mohamed Berekaa, Doaa Osama, Mohamed Farouk Ezzeldin, Nagih M. Shaalan, Abdulaziz Abdulrahman AlMulla

**Affiliations:** 1Department of Physics, College of Science, King Faisal University, P.O. Box 400, Al-Ahsa 31982, Saudi Arabia; 2Egyptian Petroleum Research Institute, Nasr City, P.O. Box 11727, Cairo 11765, Egypt; 3Department of Environmental Health, Collage of Public Health, Imam Abdulrahman Bin Faisal University (IAU), P.O. Box 1982, Dammam 31441, Saudi Arabia; 4Basic and Applied Scientific Research Center (BASRC), Imam Abdulrahman Bin Faisal University, P.O. Box 1982, Dammam 31441, Saudi Arabia; 5Physics Department, Faculty of Science, Assiut University, Assiut 71516, Egypt

**Keywords:** modified nanohybrids, nanotubes and fatty acids, ultrasonic, Zn/Al nanolayered, layered double hydroxide (LDH), antimicrobial resistance

## Abstract

Due to an escalating increase in multiple antibiotic resistance among bacteria, novel nanomaterials with antimicrobial properties are being developed to prevent infectious diseases caused by bacteria that are common in wastewater and the environment. A series of nanolayered structures and nanohybrids were prepared and modified by several methods including an ultrasonic technique, intercalation reactions of fatty acids, and carbon nanotubes, in addition to creating new phases based on zinc and aluminum. The nanomaterials prepared were used against a group of microorganisms, including *E. coli*, *S. aureus*, *Klebsiella pneumoniae* and *Pseudomonas aeruginosa*. Experimental results revealed that a nanohybrid based on carbon nanotubes and fatty acids showed significant antimicrobial activity against *E. coli*, and can be implemented in wastewater treatment. Similar behavior was observed for a nanolayered structure which was prepared using ultrasonic waves. For the other microorganisms, a nanolayered structure combined with carbon nanotubes showed a significant and clear inhibitory effect on *S. aureus*, *Klebsiella pneumoniae* and *Pseudomonas aeruginosa*. It is concluded that the nanolayered structures and nanohybrids, which can be modified at low cost with high productivity, using simple operations and straightforward to use equipment, can be considered good candidates for preventing infectious disease and inhibiting the spread of bacteria, especially those that are commonly found in wastewater and the environment.

## 1. Introduction

The increase in bacterial resistance against multiple antibiotics has attracted the attention of scientists to the development of new materials with antimicrobial properties to prevent infectious diseases and inhibit the spread of bacteria, especially those found in wastewater. Many nanomaterials, such as nano-Ag, nano-ZnO, nano-TiO_2_, and carbon nanotubes have the potential to be used in the disinfection of water and wastewater [1,2,3].

Different types of nanostructures have been generated for the purpose of water purification, including nanolayered, nanohybrid, carbon nanotube and nanocomposite materials [4]. Many studies relating to the removal, or partial removal, of organic pollutants [5], and to bacterial deactivation [6], by nanomaterials have been published.

Layered double hydroxide (LDH), which consists of nanolayered structures, has witnessed significant growth in interest in many fields due to its unique physical and chemical characteristics. As a hydrotalcite-like compound, LDH sheets hold a net positive charge which is compensated by negatively charged intercalating anions and water molecules present in the interlaminar region.

Recently, there has been growing interest in the widespread application of functionalized layered double hydroxide (LDH) (Figure 1) as a type of two-dimensional inorganic layered material, especially in catalysis, for the sorption of pollutants, in bio-sensors, for corrosion inhibition, drug delivery, photocatalytic water remediation and antimicrobial material development, and for environmental protection. In addition, the biocompatibility of LDH and its low toxicity and cost has attracted attention [7,8,9,10,11,12,13]. Due to the limited applicability of pure LDH, several crucial modifications have been introduced using appropriate organic and inorganic modifying agents. These modifications enable the enlargement of the basal spacing and increase the organophilic properties of LDH [14]. To increase the efficiency of the material, pristine LDH material has been subjected to a wide range of surface modifications, intercalation and the loading of substrates, functional groups, and structural components [14,15,16,17,18].

Some LDH and LDH composite modified nanomaterials have been widely applied as antimicrobial agents in different areas, including water purification, food packaging and wound healing [14], due particularly to the bactericidal effect of nanometals incorporated in the layered double hydroxide structure [19,20]. Recently, LDH nanomaterials have been proposed as antibiotic carriers which can facilitate the controlled release of the carried drug [21,22,23]. Abdel Moaty et al. [24] synthesized a modified Zn-Fe LDH material with antimicrobial activity against a wide range of Gram-negative and Gram-positive bacteria, especially MTSA *S. aureus*, and some fungi. As well as the bactericidal action of several modified LDH nanomaterials, significant bacteriostatic activity has also been recorded [25]. The surface of LDH material has also been modified by combination with bio-absorbent material to form composites with increasing efficiency. Olivera et al. [26] showed that surface modification of LDH by combination with proteins resulted in a broad spectrum of heavy metal sorption properties and antimicrobial, as well as antioxidant, activity.

Many fabrication methods have been used for production of a range of LDHs. The most common include co-precipitation, anion exchange, hydrothermal, urea hydrolysis, microwave, and sol-gel methods. A co-precipitation process accompanied by ultrasonic wave treatment has been described in some reports as a very promising technique for the development of innovative LDH materials with new properties [27,28,29]. The sonication process, which is considered an environmentally friendly and economic approach, has been used to produce uniform dispersion in LDH structures [30,31].

The present study involved the development of new modified LDH materials with effective antimicrobial activity against a wide range of Gram-negative and Gram-positive bacteria, especially *E. coli*, which exists in wastewater and sewage water. A series of modifications were performed to develop LDH nanolayered structures. An ultrasonic technique was used to modify the size of platelets and the LDH layers. Organic species were intercalated inside the LDHs to control the interlayered spacing of the nanolayered structures by formation of organic-inorganic nanohybrids. In addition, carbon nanotubes were used as a filler for the LDHs to increase the antibacterial performance of the nanolayered LDH structures. The nanolayered structure of the LDHs was modified by changing the percentage of di- and tri-valent inorganic metals. By applying these modifications, a series of nanolayered structures and nanohybrids were prepared which were assessed for potential antimicrobial activity against a group of microorganisms that are commonly found in the environment, specifically, *E. coli*, *S. aureus*, *Klebsiella pneumoniae* and *Pseudomonas aeruginosa*.

## 2. Materials and Methods

### 2.1. Preparation of Nanolayered Structures and Nanohybrids

Five samples of Al/Zn nanolayered materials were prepared and modified by different techniques. The standard sample was prepared by reacting an aqueous solution of zinc nitrate (0.07 mol) with aluminum nitrate in the presence of urea. It was coded as ZA-1. The percentage of aluminum was 30 wt.%. A sample, ZA-2, was prepared and modified by decreasing the percentage of aluminum to 15 wt.%. A sample ZA-3 was prepared by the same procedure as for the standard sample and modified using an ultrasonic technique. An ultrasonic technique was used to modify the precipitation process of the product. In the presence of carbon nanotubes (CNTs), the fourth sample was formed and coded as ZA-4. CNTs were used as seeds for building the Al/Zn nanolayered structure. The fifth sample, ZA-5, was modified by intercalating organic species to build an organic-inorganic nanohybrid. An aqueous solution of a stearic acid sodium salt and carbon nanotubes were used as additives during the precipitation of the nanohybrid. Urea and polyvinyl alcohol were used as a pH-controller and a binder; respectively. The temperature of the reaction in all preparations was kept at 90 °C. The products were washed by distilled water. They were dehydrated at room temperature for 48 h.

### 2.2. Characterization Techniques

X-ray diffraction (XRD) is considered one of the main tools for determining nanolayered structures. X-ray diffraction (Bruker-AXS, Bruker Company, Karlsruhe, Germany) was used with wide angle X-ray scattering over 2θ = 4 to 50°, in steps of 0.1 or 0.02°, with Cu-Kα radiation (λ = 0.154 nm). Transmission electron microscopy (TEM) was used to observe the shape and size of the LDHs.

The measurements were performed at room temperature. The different elements of the prepared materials were identified by energy-dispersive X-ray spectroscopy (EDX) using an electron probe micro-analyzer JED 2300 (JEOL Company, Tokyo, Japan). Applying molecular vibrational spectroscopic techniques, Fourier transform infrared spectroscopy (FTIR) was used to determine the functional groups of the prepared materials using a Perkin Elmer Spectrum 400 machine (Perkin Elmer Company, Waltham, MA, USA). Thermal analyses, consisting of thermal gravimetric analysis (TGA) and differential scanning calorimetry (DSC), were performed. TGA was carried out using a TA thermogravimetric analyzer (series Q500) (TA Company, New Castle, PA, USA) to characterize the decomposition and thermal stability of the materials under a nitrogen atmosphere. Using a TA series Q 600 (TA Company, New Castle, PA, USA), DSC analysis was performed under a flow of inert gas with a heating rate of 10 °C min^−1^.

### 2.3. Bacterial Strain

To test the potential antimicrobial effect of surface-modified Zn-Al LDH nanomaterial, a group of bacterial strains was used, including *E. coli, S. aureus, Klebsiella pneumoniae* and *Pseudomonas aeruginosa,* which were obtained from the culture collection, College of Medicine, University of Imam Abdulrahman Bin Faisal (University of Dammam, Dammam, Saudi Arabia), Saudi Arabia.

### 2.4. Microbiological Media

Two types of microbiological media were used: nutrient agar and diagnostic sensitivity test agar medium for bacteria. The media composition and preparation were as follows:

### 2.5. Nutrient Agar (OXOID, England)

Nutrient agar medium (OXOID, Hampshire, England) was used to support the growth of all bacterial candidates. The typical formula (g/L) composition was: lab-lemco powder 1.0; yeast extract 2.0; peptone 5.0; sodium chloride 5.0; and agar 15.0. A specific amount of the medium was prepared and sterilized according to the manufacturer’s instructions.

### 2.6. Diagnostic Sensitivity Test Agar Medium (D.S.T. Agar) (HIMEDIA, Thane West, India)

To ensure better diffusion of the material, a ready prepared diagnostic sensitivity test agar medium was used. The typical formula (g/L) was: proteose peptone 10.0; veal infusion solids 10.0; dextrose 2.0; sodium chloride 3.0; disodium phosphate 2.0; sodium acetate 1.0; adenosine sulphate 0.01; guanine hydrochloride 0.01; uracil 0.01; xanthine 0.01; aneurine 0.0002 and agar 15.0. This type of medium can support the growth of most bacteria, even fastidious pathogens. A specific amount of medium was prepared and sterilized according to the manufacturer’s instructions.

After preparation, the medium was sterilized by autoclaving at 121 °C for 15 min. The medium was subsequently poured onto Petri plates, left to solidify at room temperature and stored in a refrigerator until use.

### 2.7. Antimicrobial Inhibitory Test (Agar Diffusion Method)

To test the possible antimicrobial activity of the nanocomposite materials, a stock solution of known sample concentration was prepared as follows: A known weight of the powder material was sterilized by exposure to UV radiation for 3 min. The powdered material was subsequently dissolved in water (5–25 mg/mL) or DMSO (10 mg/mL). To test the antimicrobial activity of the nanocomposite material a modified agar diffusion method was used [32]. In this process, the bacterial strains were spread on the surface of the diagnostic sensitivity test agar medium using a sterile cotton swab. Afterwards, pores were made (6 mm in diameter) using a cork porer, and different concentrations of nanocomposite sample were placed in these. Subsequently, plates were incubated at 37 °C for 24 h. At the end of the incubation period, the plates were examined for the appearance of an inhibition zone that was measured to an accuracy of 0.5 mm in two perpendicular locations. The results were expressed as the mean zone of inhibition (in mm ± standard deviation) beyond the standard well diameter (6 mm).

## 3. Results

The nanolayered structures and nanohybrids were prepared and modified to produce five samples ZA-1, ZA-2, ZA-3, ZA-4, and ZA-5. Table 1 summarizes the effect of modification on the d-spacing of the nanolayered structures.

### 3.1. Inhibitory Effect of Different Modified Al/Zn Nanolayered Structures

X-ray diffraction (XRD) was used to identify the layered structure of the prepared Al/Zn LDHs. The XRD diagram of the sample ZA-1 conformed to the layered structure of the natural hydrotalcite (JCPDS file No. 37–629) and the synthetic Zn–Al LDH (JCPDS file No. 48–1022), as shown in Figure 2a.

A sharp peak was observed for the main plane (003) at d = 0.76 nm, as shown in Figure 2a. The XRD diagram exhibited other peaks of the planes (006) and (009) at d = 0.38 nm and d = 0.26 nm, respectively. In addition, Figure 2a shows that there was a clear relationship between the values of the basal planes, as follows: d_(003)_ = 2 × d_(006)_ = 3 × d_(009)_. This implies that the nanolayered structure of ZA-1 included highly ordered nanolayers along axis “c”. The value “c” represents the thickness of the brucite-like layer and the interlayer distance. Depending on the interlayered spacing of plane (003), the value “c” was assessed by 3 × d_(300)_ to equal 2.28 nm. This precisely matched the value “c” of hydrotalcite, 2.28 nm.

The second sample, ZA-2, was formed by reducing the percentage of aluminum. The XRD pattern of sample ZA-2 indicated that the intensity of all the reflections of the LDHs decreased without shifting of the parameter ©, as shown in Figure 1. The low crystallinity of the ZA-2 sample could be related to the relatively large zinc content in the sample, thus producing new phase zinc hydroxyl carbonate, which is consistent with the peak observed at 0.67 nm. This implies that the ZA-2 sample had a lower percentage of nanolayered structures than that of ZA-1.

Previous studies have found that natural samples of layered double hydroxides (pyroaurite) have a plate-like morphology. In addition, the dimensions of these plates are in millimeters and not at a nano scale [33]. It is known that the morphology of the hydrotalcite, when carefully crystallized, is hexagonal platy [34]. A similar morphology was observed for ZA-1 with the dimensions of the plates of the order of 100 nm, as shown in Figure 2b. Figure 2b shows that the thickness of the plates was 10 nm. Energy-dispersive X-ray spectrometry analysis revealed that the inorganic elements zinc and aluminum occurred in the platelets of sample ZA-1, as shown in Figure 2b inset. The interlayered anions were confirmed by signals of carbon and oxygen at low energy. It should be noted that the Cu signal observed in the spectrum was attributed to the copper grid of the TEM testing setup.

For sample ZA-2, which had a low concentration of aluminum, the TEM images showed irregular plates and shapes, as shown in Figure 2c, consistent with the XRD results. The low concentration of aluminum in sample ZA-2 was confirmed by the intensity of the aluminum peaks in the EDX spectra, as shown in the Figure 2c inset.

Thermal measurements using thermogravimetric analysis, TGA, and differential scanning calorimetry, DSC were obtained to confirm nanolayered structure formation and to characterize the thermal behavior of the internal contents of sample ZA-1. The recorded thermogravimetric and differential scanning calorimetric diagrams are shown in Figure 2d. Thermal degradation of the internal contents of ZA-1 was observed to occur in several stages with various mass rate losses, depending on the nature of the interlayer anions. As can be seen from the TG curves of ZA-1, the degradation process exhibited four weight losses, as shown in Figure 2d. The first mass loss (12 wt.%) was detected at 170 °C which contributed to the loss of both the physisorbed and the interlayer water. The other three mass losses (16 wt.%) were observed at higher temperatures (170–310 °C). These event transitions can be attributed to the decomposition of the interlayered anions in addition to an ihydroxylation process. The results obtained imply that the nanolayered structure of ZA-1 contained two different anions with two different mass rate losses. This finding was confirmed by the DSC curve. The DSC curve of AZ-1 showed three endothermic peaks at 160 °C, 211 °C and 276 °C, consistent with the removal of the intercalated water and the decomposition of the intercalated cyanate anions, in addition to the degradation of the intercalated carbonate anions, as shown in Figure 2d.

By comparison with ZA-1, the TG curve for ZA-2 was a little different because the amount of the interlayered water decreased to 5 wt.% observed at 161 °C, indicating a reduction in the nanolayered structure of the LDHs. The total weight loss was 26 wt.%. Decomposition of carbonate and cyanate anions was observed in the temperature range 161–310 °C, as shown in Figure 2d.

To determine the antimicrobial activity of the prepared materials, ZA-1 and ZA-2, 100 µL of the test samples was deposited in defined pores after cultivation of bacteria on the surface of the media. After incubation, the inhibitory effect was recorded, and the approximate diameter of the inhibition zone was determined. The results shown in Figure 3 and presented in Table 2 indicated that ZA-1 had an inhibitory effect on all tested bacterial strains, such as *Klebsiella pneumoniae, E. coli* and *P. aeruginosa*, especially when dissolved in DMSO. Lower concentrations of the material had no effect on the strains tested (Figure 3).

The major inhibitory effect occurred for *Pseudomonas aeruginosa*, followed by *E. coli* and *Klebsiella pneumoniae* (15, 8.5 and 8.1 mm, respectively). Consistent with this study, Moaty et al. [24] observed that LDH exhibited long-lasting antibacterial activity against Gram-negative bacteria including *K. pneumoniae, E. coli*, and *P. aeruginosa*, as well as some Gram-positive bacteria, including *S. aureus*.

The antibacterial effect of the nanolayered structure of ZA-1 can be explained by the adhesion of bacteria to the surfaces of the nanolayers of ZA-1 via van der Waals interaction and electrostatic exchanges [35]. The nanolayers were able to successfully stop or inhibit the division and growth of the bacteria. It has been suggested that DNA loses its replication capability and cellular proteins are deactivated under these conditions [36].

Modification of the LDH preparation using Zn/AL nitrate 15% (ZA-2) showed variable inhibitory effects on the activity of the nanomaterial. A limited inhibitory effect was observed against *S. aureus* and a clear effect was observed against the Gram-negative bacterium *E. coli* when dissolved on water or DMSO, with inhibitory effects of approximately 10 mm and 11.3 mm, respectively.

The results imply that the nanolayered structure of LDH has an important role in its antimicrobial activity as the ZA-2 sample modified the nanolayered structure by reducing the percentage of aluminum, leading to the formation of another structure of zinc hydroxyl carbonate. Therefore, a variable inhibitory effect was observed for the antimicrobial activity of ZA-2.

### 3.2. Effect of Ultrasonic Technique on the Inhibitory Effect of the Nanolayered Structures Al/Zn

According to many reports [37,38,39], ultrasound energy can cause physical and chemical changes to the structures of materials during their preparation because of the breakdown of cavitation bubbles in the liquid system. In our study, an ultrasonic technique was used to modify and develop the size and nanolayered structure of Al/Zn LDHs in the preparation of sample ZA-3. X-ray diffraction of ZA-3 showed sharp peaks at low angles and weak peaks at high angles, consistent with the nanolayered structures shown in Figure 4a. The sharp peaks were observed at 2θ = 11.71° and 23.41°, consistent with d-spacing at 0.754 nm and 0.379 nm: respectively. The weak peaks occurred at 2θ = 34.60°, 39.16° and 47.21°, consistent with d-spacing at 0.253 nm, 0.230 nm and 0.195 nm, respectively. Figure 4 shows that the first three symmetric peaks, which were due to the basal (003), (006) and (009) planes, reflect the good arrangement of the successive diffraction of their planes, i.e., 0.754 nm = 2 × 0.379 nm = 3 × 0.253 nm. The two asymmetric peaks observed at 0.230 nm and 0.195 nm were associated with the non-basal (015) and (018) planes. By comparison with the hydrotalcite-like material (JCPDS file No. 37–629) and the synthetic Zn–Al LDH (JCPDS file No. 48–1022), the ZA-3 sample exhibited the nanolayered structure of zinc aluminum hydroxide carbonate hydrate.

Figure 4a shows a weak peak appearing at 2θ = 19.4° (d-spacing = 0.456 nm) corresponding to the reflection of the 101 plane of polyvinyl alcohol which was used as a binder during the building of the nanolayered structure [40]. Very weak peaks were observed at 2θ = 12.96°, 26.79° and 36.80° consistent with the hydrozincite phase Zn_5_(OH)_6_(CO_3_)_2_ (JCPDS file No. 72–1100). The results imply that the nanolayered structure of ZA-3 contained traces of the hydrozincite phase.

The crystallite sizes of the particles of ZA-3, which were assessed from the XRD peak widths of (003), (006), (009), and (015), were calculated to be 50 nm, 42 nm, 65 nm and 40 nm, respectively. This means that the average particle size of the ZA-3 sample was 49 nm. TEM images confirmed the nanoscale of the sample ZA-3, as shown in Figure 4b,c. Figure 4b shows individual plates with thickness less than 50 nm. Figure 4c shows that the shape of the particles had the appearance of tapes, with a width of 25 nm, as indicated by arrows in the TEM images. The components of the nanolayers were identified by energy-dispersive X-ray spectrometry (EDX). Zinc, aluminum, oxygen, and carbon were observed in the EDX spectrum, as seen in Figure 4b (inset).

The details of the interlayered region of ZA-3 were determined by thermal analyses. The thermal characteristics were determined by thermogravimetric analysis (TGA) and differential scanning calorimetry (DSC). The TG curve showed that 32 wt.% of ZA-3 was lost after heating to 800 °C occurring in three stages. The first stage was 8 wt.% associated with removal of the interlayered water, and was accomplished at 173 °C. The second stage occurred at 316 °C associated with removal of 18 wt.% of the interlayered anions. During the third stage, 6 wt.% was lost through dehydroxylation of the nanolayers. The DSC curve confirmed these three stages showing three endothermic peaks at 153 °C, 211 °C and 279 °C, as shown in Figure 4d.

The results of X-ray diffraction, TEM, EDX and thermal analyses indicated that the ZA-3 sample consisted of nanotapes with nanolayers of 0.48 nm thickness connected by pillars of carbonate anions concentrated in the region 0.274 nm between the nanolayers.

By comparison with the standard sample ZA-1, the nanolayered structure of ZA-3 could be more effective in inhibiting bacteria because the ZA-3 sample structure is of lower nanosize than that of the ZA-1 sample. According to this trend, the ZA-3 sample was tested for its antimicrobial activity (Table 3, Figure 5 and Figure 8).

The ZA-3 sample resulted in a major inhibitory effect on the Gram-negative bacteria *E. coli* (18.5 mm), followed by *K. pneumoniae* (14.6 mm) and *P. aeruginosa* (11.5 mm), with no effect on the Gram-positive bacterium *S. aureus*.

The results suggest that Gram-negative organisms are relatively more sensitive towards the ZA-3 sample than Gram-positive organisms. The difference in the observed antibacterial activity between Gram-negative and Gram-positive bacteria can most probably be explained by the different structures and the chemical composition of the cell surfaces of the two bacterial types because the inhibitory effect of the nanolayers depends on the interaction between the surface of the cells and the nanolayers. It is known that Gram-positive bacteria possess a thick outer peptidoglycan layer, while Gram-negative bacteria have a thin outer phospholipidic membrane. The outer membrane of the Gram-negative bacterial cell wall has special proteins called porins which form pores through the outer membrane and allow hydrophilic molecules to diffuse into the periplasm. Therefore, the hydrophilic nanoplates of ZA-3 have the ability to diffuse inside the periplasm and cause damage to Gram-negative cells. This is not possible for the Gram-positive bacteria because they have thick and rigid peptidoglycan layers for protection.

By comparison with the original sample ZA-1, the activity of the sample ZA-3 showed severe antimicrobial effects against *E. coli*, which is a common microbial representative in wastewater and sewage water. The inhibitory effect of ZA-3 was more than two times higher than that of ZA-1. This implies that the ZA-3 sample can be considered a good candidate for treating wastewater and sewage water.

### 3.3. Effect of Carbon Nanotubes on the Inhibitory Effect of the Nanolayered Structures Al/Zn

Much research effort has been directed towards the development of nanostructures in combination with carbon nanotubes. Therefore, functional carbon nanotubes were used as an additive during the building of the Al/Zn nanolayered structures to produce the ZA-4 sample. At the same time, polyvinyl alcohol was used as a binder for the nanolayered structure.

The ZA-4 sample was characterized using different techniques to identify the texture of its nanolayered structures. Figure 6a shows the X-ray diffraction pattern obtained. It indicates that the ZA-4 sample consisted of two nanolayered structures. A first structure was evident by the observation of three peaks at 2θ = 11.79°, 23.4° and 33.35° which are consistent with d-spacing of 0.75 nm, 0.376 nm and 0.268 nm, respectively. The good arrangement and similarity between 0.75 nm, double 0.376 and triple 0.268 nm indicates a nanolayered structure consistent with the hydrotalcite-like material (JCPDS file No. 37–629) and the synthetic Zn–Al LDH (JCPDS file No. 48–1022). This implies that the first structure of the ZA-4 sample was a nanolayered structure of zinc aluminum hydroxide carbonate hydrate.

A second nanolayered structure was indicated by the peaks observed at 2θ = 12.99°, 26.74° and 36.60°, which are consistent with d-spacing of 0.681 nm, 0.333 nm and 0.245 nm, respectively, as shown in Figure 6a. The similarity among the d-spacing of the main peak 0.681 nm, the double of the second peak (2 × 0.333 nm) and the triple of the third peak (3 × 0.245 nm) confirmed the construction of a new nanolayered structure by combination with carbon nanotubes and polyvinyl alcohol. According to the XRD calculations, the second nanolayered structure represented 50% of the ZA-4 sample.

Figure 6a shows a strong and sharp peak at 2θ = 19.50, consistent with d-spacing of 0.445 nm. This peak is characteristic of polyvinyl alcohol and is due to reflection of the plane of (101) [41].

It is known that pure polyvinyl alcohol has a semi-crystalline structure [42,43]. The crystalline nature of PVA is due to the hydrogen bonding of the hydroxyl groups of PVA causing small crystallites to form in the amorphous PVA matrix related to the PVA semi-crystalline structure [44].

The results imply that combination with the nanolayered structure converted PVA from semi-crystalline material to crystalline material by increasing the H-bonds of the chains of PVA with the nanolayers of Al/Zn LDHs and CNTs. By evaluating the XRD results, we concluded that the addition of CNTs with PVA during preparation of the sample A-4 produced two nanolayered structures. The first structure was the normal nanolayered structure of Al/Zn LDHs. The other structure involved combination of the PVA chains and carbon nanotubes via hydrogen bonds, which decreased the interlayered spacing of the nanolayered structure from 0.750 nm to 0.681 nm.

TEM images confirmed the presence of two nanolayered structures. Figure 6b shows clear plates with irregular shapes for the normal nanolayered structure of Al/Zn LDHs. A second nanolayered structure, modified by CNTs and PVA, was observed in the TEM images, as shown and marked in Figure 6b by arrows. EDX analysis confirmed the presence of inorganic species of zinc, aluminum and oxygen, as shown in Figure 6b (inset).

Thermal analyses confirmed the nanolayered structures using TG curves and DSC diagrams (Figure 6c). The TG curve showed that 28% of the sample was lost after heating to 305 °C. This implies that 28 % of the content of the sample, which consisted of water and anions, was confined to the interlayered region. In addition, the hydroxyl groups of the nanolayers, which represented 6% of the sample content, were removed by heating to 600 °C. The DSC diagram showed two endothermic peaks and one exothermic peak, confirming the TG results. The two endothermic peaks were observed at 139 °C and 275 °C, consistent with the removal of the interlayered water and anions, respectively. The exothermic peak was due to a crystallization process after removal of the hydroxyl groups.

After identifying the structure of the ZA-4 sample, its antimicrobial effects on all tested bacterial strains were evaluated and compared with those for sample ZA-1 (Table 4, Figure 3, Figure 5 and Figure 8a).

The ZA-4 sample showed high antimicrobial activity against Gram-negative bacteria *P. aeruginosa* and *K. pneumoniae* (approximately 16.5 and 8.4 mm, respectively) and the Gram-positive bacterium *S. aureus*.

By comparison with the major inhibitory effect of the basal nanolayered structure ZA-1, incorporation of CNTs enhanced the lethal effect of LDH nanocomposite material. Generally, the lethal effect of CNTs is due to bacterial cell membrane damage and generation of toxic reactive oxygen species (ROS) [45,46]. Rajavel et al. [47] reported that the lethal effect of CNTs against *K. pneumoniae* and *P. aeruginosa* was due to ROS production and partial penetration of the nanotubes into the cells, affecting cell membrane fluidity [48,49].

*P. aeruginosa* and *K. pneumoniae* are well known for biofilm formation. This biofilm could be a major target for Zn/AL nitrate 30%, leading to inhibition of Gram-negative and Gram-positive bacterial growth. Yin, et al. [50] recorded a characteristic inhibitory effect of Mg/Al-LDH on biofilm formation by *Vibrio parahaemolyticus* due to destruction of macromolecules, such as polysaccharides and DNA, during biofilm formation. Moreover, the modified LDH material prevented the gelation of polysaccharides and no effect was observed on proteins (suggesting it could be used as a drug delivery vehicle). Moreover, a similar effect of MWCNTs on biofilm formation by *Klebsiella oxytoca* and *Pseudomonas aeruginosa* was recorded by Malek et al. [51], and of SWCNTs on *Bacillus anthraces* biofilm formation [52].

### 3.4. Effect of Nanohybrids on the Inhibitory Effect of Al/Zn Nanolayered Structures

To expand and widen the entrance gate of the nanolayered structures, fatty aliphatic acids with long chains were used to increase the interlayered spacing by building organic-inorganic nanohybrids using intercalation reactions. The ZA-5 sample was prepared by intercalating stearic acid (C18) inside the nanolayered structure, and was then characterized by X-ray diffraction, TEM, EDX, thermal analyses and FTIR.

Figure 7a shows the X-ray diffraction pattern of the ZA-5 sample. This showed three clear peaks at low 2θ = 4.1°, 6.2°, and 8.3°. These peaks indicate that a new nanolayered structure was formed after the intercalation reactions. In this nanolayered structure, the interlayered spacing increased to 1.6 nm, 1.4 nm and 1.06 nm. The normal nanolayered structure of Al/Zn LDHs was observed in the XRD pattern. Figure 7a shows the main characteristic peaks of Al/Zn LDHs at 11.6°, 23.4° and 34.5°, consistent with d-spacing of 0.76 nm, 0.38 nm and 0.26 nm, respectively. The three peaks are related to the reflections of the (003), (006), and (009) planes, in agreement with the standard data for zinc aluminum hydroxide carbonate hydrate (JCPDS no. 48–1022). The planes (015) and (018) were observed at 0.23 nm and 0.19 nm, confirming the nanolayered structure of Al/Zn LDHs. In addition, Figure 7a shows a broad peak at 0.45 nm, confirming the presence of polyvinyl alcohol. According to the XRD results, the ZA-5 sample consisted of two different nanolayered structures. The first structure is a nanohybrid with interlayered spacing of more than 1.5 nm. The second structure is a normal nanolayered structure with an interlayered spacing of 0.75 nm. Figure 7b shows the TEM image and the EDX spectrum for the ZA-5 sample. These exhibit large plates associated with the main elements of zinc, aluminum and oxygen in the nanolayers.

FTIR confirmed the presence of a nanohybrid with sharp bands at 2959 cm^−1^, 2917 cm^−1^ and 2853 cm^−1^. These bands were due to C–H stretching absorption of the organic species. A bending band of C–H was also observed at 1462 cm^−1^, as shown in Figure 7c. In addition, asymmetric and symmetric stretching vibrations of carboxylate were observed at 1392 cm^−1^ and 1538 cm^−1^, respectively. The hydroxyl groups of the nanolayers of the ZA-5 sample were reflected in two bands at 3580 cm^−1^ and 3280 cm^−1^. This implies that there were two types of hydroxyl group, confirming the presence of two nanolayered structures. The first type of hydroxyl group was due to the normal nanolayered structure of Al/Zn LDHs. The second type of hydroxyl groups was affected by hydrocarbon chains belonging to the nanolayered structure of the Al/Zn nanohybrids.

Thermal analyses confirmed the formation of a nanohybrid by the TG curve and DSC diagram, as shown in Figure 7d. The TG curve indicated that 55 wt.% was lost by heating to 600 °C. This implies that more than 50% of the structure of the ZA-5 sample was concentrated in the interlayered region as water and anions, confirming the formation of a nanohybrid structure. The DSC diagram confirmed the TG data with a broad and large exothermic peak at 476 °C, indicating oxidation reactions of the hydrocarbon species. The DSC diagram also showed two small endothermic peaks at 131 °C and 279 °C, indicating the presence of the normal nanolayered structure of Al/Zn LDHs as a secondary phase. From these analyses, it can be concluded that the ZA-5 sample had two nanolayered structures with two interlayered spacings, one below 1 nm and the other above 1.5 nm.

To evaluate the effect of organic species on the antimicrobial activity of the nanolayered structures, the ZA-5 sample was used against the Gram-negative bacteria *E. coli*, *P. aeruginosa* and *K. pneumoniae*, as well as the Gram-positive bacterium *S. aureus* (Table 5, Figure 8a,b).

Finally, incorporation of stearic acid during preparation of the modified nanomaterial ZA-5 (HF5 Zn/AL nitrate + CNTs + stearic acid) maintained the activity of LDH plus CNTs (ZA-4) for Gram-negative bacteria, especially *E. coli*, *P. aeruginosa* and *K. pneumoniae* (with inhibitory effects of 16.7, 13.7 mm and 12.5 mm, respectively); no effect was recorded against *S. aureus*.

## 4. Discussion

The synthesis of novel antimicrobial nanolayered structures and nanohybrid materials against a wide range of Gram-positive and Gram-negative bacteria in this study represents a major advance in the prevention of the spread of infectious diseases in the environment. Fortunately, most of the nanocomposite materials produced showed an enhanced antimicrobial effect against Gram-positive and Gram-negative bacteria. The observed in vitro antimicrobial activity of surface-modified Zn-Al LDH nanocomposite materials against *E. coli* indicates the potential for their use in the bioremediation of sewage wastewater. Consistent with the current study, Zn-Fe layered double hydroxide (LDH), with nitrate as the interlayer anion, has been synthesized and showed antimicrobial effects against a wide range of microbes including *E. coli* [24]. Moreover, Mengxue et al. [53] showed that ZnO-dotted nanohybrids, hydrothermally derived from Zn Al-layered double hydroxides, expressed potent and durable antimicrobial activity. The bactericidal effect of Zn-Al LDH nanocomposite material is probably due to the adsorption of negatively charged bacterial cells to the positively charged modified Zn-Al LDH material, and the toxic effect of hydroxyl ions, radicals, and reactive oxygen species generated on the bacterial cell components, including the cell membrane, proteins, and the DNA genetic material [24,54]. Moreover, the Zn and Al ions enter the bacterial cells and interact with the cell membrane and genetic material via electrostatic attraction and denature cell proteins [13,53,55].

Awassa et al. [56] suggested that the partial release of metallic ions in aqueous dispersion could be the major reason for the antimicrobial activity of modified LDHs, e.g., ZnII-AlIII LDH against *E. coli* and *S. aureus*, and represent a solution to the problem of antibiotic resistance in many bacteria. Interestingly, the inhibitory effect was found to be significantly affected by the ZnII:AlIII molar ratio and the exchange of carbonate anions with other anions, and by increased ZnII ion concentration (with the inhibitory concentration decreasing from 12 to 0.375 mg·mL^−1^).

Remarkably, Peng et al. [57] found that ZnII-based LDHs exhibited enhanced antimicrobial activity when compared with MgII-based LDHs and even CuII-based LDHs [11]. Furthermore, the antimicrobial effect of zinc ions in ZnII-based LDHs has been suggested to be the major reason for their antimicrobial activity [58].

The successful intercalation of organic species to build organic-inorganic nanohybrid materials using an aqueous solution of stearic acid sodium salt and carbon nanotubes (CNTs) during the precipitation of the nanohybrid resulted in a significant increase in the material’s antibacterial activity (ZA-5). The involvement of carbon nanotubes (CNTs) in surface modification to form Zn-Al/LDH nanocomposite material enhanced antimicrobial activity. The antimicrobial activity of CNTs against Gram-negative bacteria, including *E. coli*, resulted from disruption of cellular membrane integrity, damage to the cell wall, decrease in metabolic activity, cell lysis and leakage of cell constituents [59,60,61,62]. Finally, it can be concluded that these nanohybrids showed encouraging results in relation to tackling bacterial resistance against multiple antibiotics. The results obtained may attract the interest of scientists in the potential synthesis of nanolayered structures with antimicrobial properties able to prevent infectious diseases and inhibit the spread of bacteria, especially those related to the environment and wastewater.

## 5. Conclusions

The present study focused on developing and modifying nanolayered structures of Al/Zn LDH intended to be effective for overcoming infectious diseases and inhibiting the spread of bacteria, especially those commonly found in the environment and wastewater, by studying their antibacterial activity against a group of microorganisms, including *E. coli*, *S. aureus*, *Klebsiella pneumoniae* and *Pseudomonas aeruginosa*.

Utilizing carbon nanotubes and fatty acids, a nanohybrid was prepared which showed antimicrobial activity against *E. coli*, which can be applied to wastewater treatment. Similar activity against *E. coli* was observed for a nanolayered structure which was prepared using ultrasonic waves. For the other microorganisms examined, a nanolayered structure combined with carbon nanotubes showed a significant and clear inhibitory effect on *S. aureus, Klebsiella pneumonia* and *Pseudomonas aeruginosa*. It is concluded that the nanolayered structures and nanohybrids synthesized, which can be modified with high productivity at low cost, using simple operations and easily used equipment, represent good candidates for overcoming infectious diseases and inhibiting the spread of bacteria, especially those that are commonly found in the environment and wastewater.

## Figures and Tables

**Figure 1 nanomaterials-12-02749-f001:**
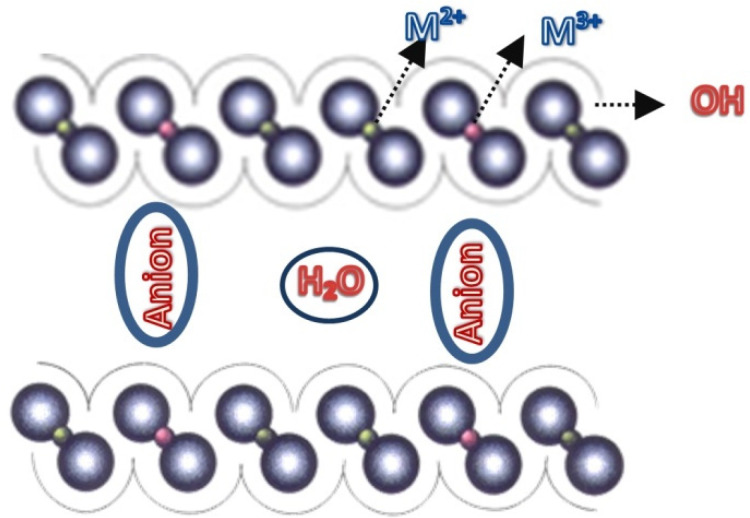
Schematic representation of LDHs.

**Figure 2 nanomaterials-12-02749-f002:**
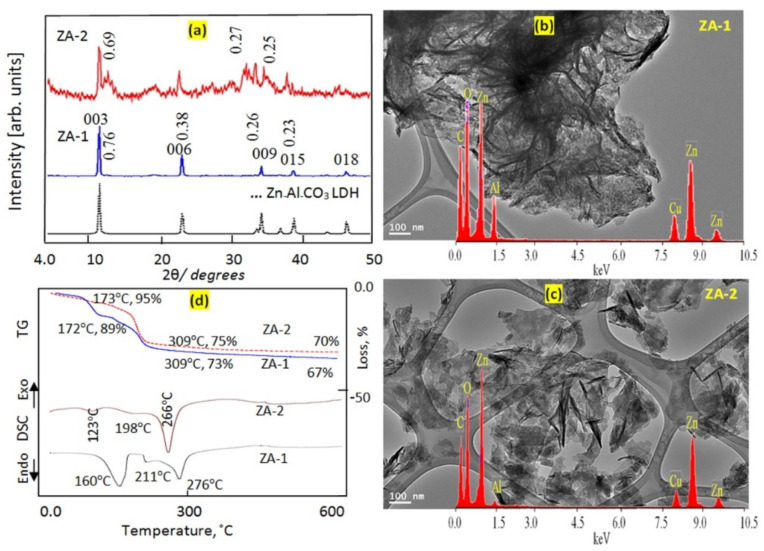
The samples ZA-1 and ZA-2: (**a**) X-ray diffraction, (**b**,**c**) TEM images and EDX spectra and (**d**) Thermal analyses.

**Figure 3 nanomaterials-12-02749-f003:**
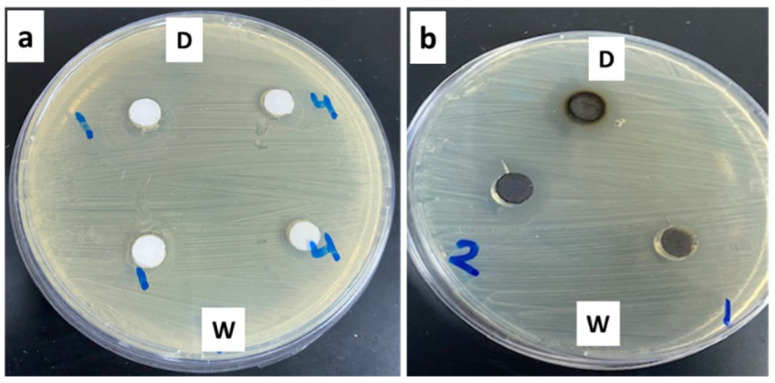
Effect of differently modified nanolayered materials and nanohybrids on growth of *S. aureus* bacterium. 1: LDH 30%, 4: LDH 30% + CNTs (dissolved in water “W” and DMSO “D”) (**a**) 1: LDH 30%, 2: LDH 15% (water) and 2: LDH 15% (DMSO) (**b**).

**Figure 4 nanomaterials-12-02749-f004:**
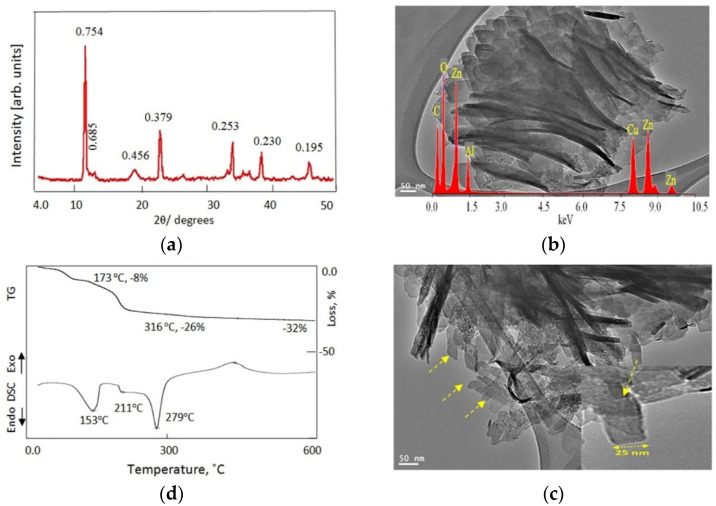
The sample ZA-3: (**a**) X-ray diffraction (numbers represent d-spacing), (**b**) TEM images, (**c**) TEM image (nano-tapes marked by yellow arrows) and EDX spectra, and (**d**) thermal analyses.

**Figure 5 nanomaterials-12-02749-f005:**
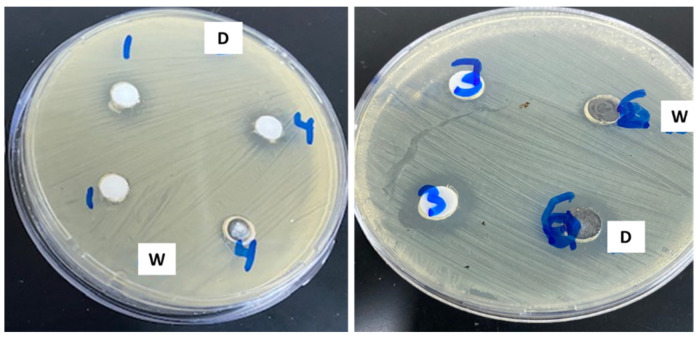
Effect of different modified nanolayered materials and nanohybrids on growth of *K. pneumoniae* bacterium. 1: LDH 30% and 4: LDH 30% + CNTs (dissolved in water “W” and DMSO “D”, respectively), 3: LDH 30% ultrasonic and 6: LDH 30% + CNTs and stearic acid (dissolved in water “W” and DMSO “D”, respectively).

**Figure 6 nanomaterials-12-02749-f006:**
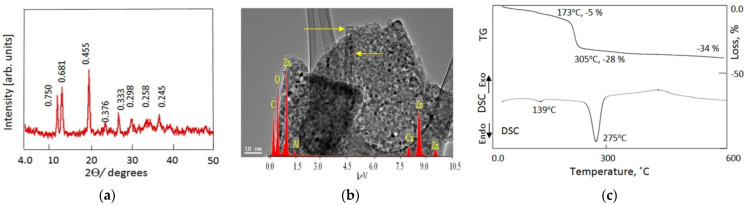
The sample ZA-4: (**a**) X-ray diffraction (numbers represent d-spacing), (**b**) TEM images and EDX spectra and (**c**) thermal analyses.

**Figure 7 nanomaterials-12-02749-f007:**
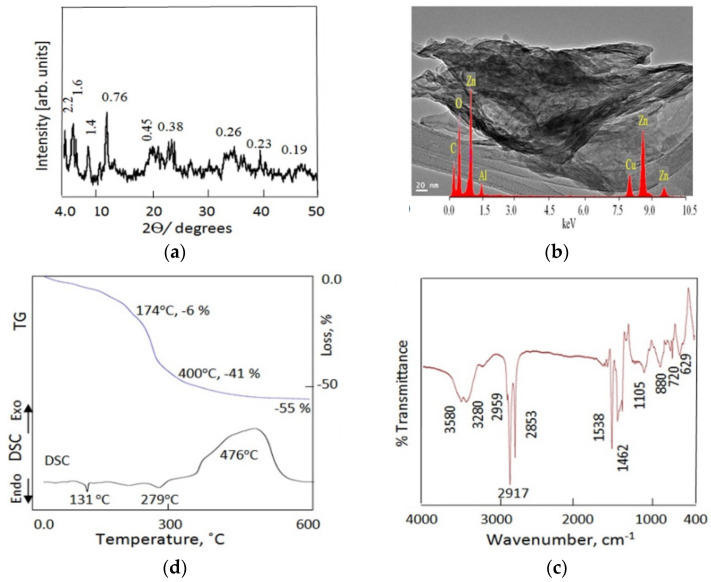
The sample ZA-5: (**a**) X-ray diffraction (numbers represent d-spacing), (**b**) TEM images and EDX spectra, (**c**) FTIR and (**d**) thermal analyses.

**Figure 8 nanomaterials-12-02749-f008:**
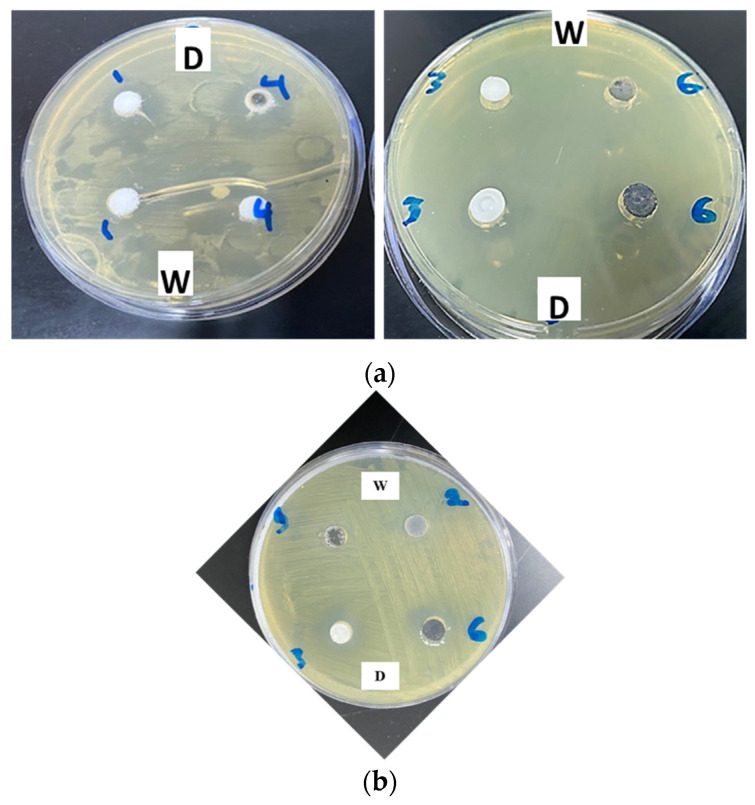
(**a**) Effect of different modified nanolayered materials and nanohybrids on growth of *P. aeruginosa* bacterium. 1: LDH 30%; 4: LDH 30% + CNTs; 3: LDH 30% (ultrasonic); and 6: LDH 30% + CNTs and stearic acid (dissolved in water “w” and DMSO “D”, respectively). (**b**) Effect of different modified nanolayered materials and nanohybrids on growth of *E. coli* bacterium. 1: LDH 30%; 2: LDH 15%; 3: LDH 30% ultrasonic (dissolved in water “w”); and 6: LDH 30% + CNTs and stearic acid (dissolved in DMSO “D”).

**Table 1 nanomaterials-12-02749-t001:** The prepared and modified nanolayered structures.

Samples	Nanolayered Structures	Nanohybrids	d-Spacing
ZA-1	√	×	0.760 nm
ZA-2	√	×	0.760 nm
ZA-3	√	×	0.754 nm
ZA-4	√	√	0.750 nm and 0.681 nm
ZA-5	√	√	2.2 nm, 1.6 nm and 1.4 nm

**Table 2 nanomaterials-12-02749-t002:** The antimicrobial activity of the prepared materials ZA-1 and ZA-2.

Table	Diameter of Inhibition Zone (mm)			
	Sample ZA-1		Sample ZA-2	
	D	W	D	W
*K. pneumonae*-1	8.1	NA	NA	NA
*S. aureus*	NA	NA	7.3	7
*E. coli*	8.5	NA	11.3	10
*P. aeruginosa*	15	NA	NA	NA

D: Zn-Al/LDH in DMSO, W: Zn-Al/LDH in water, NA: no inhibitory effect.

**Table 3 nanomaterials-12-02749-t003:** Comparison between the antimicrobial activity of the prepared materials ZA-1 and ZA-3.

Types of Bacteria	Diameter of Inhibition Zone (mm)			
	Sample ZA-1		Sample ZA-3	
	D	W	D	W
*K. pneumonae*-1	7.5	NA	14.6	NA
*S. aureus*	NA	NA	NA	NA
*E. coli*	8.5	NA	18.5	NA
*P. aeruginosa*	15	NA	11.5	NA

**Table 4 nanomaterials-12-02749-t004:** Comparison between the antimicrobial activity of the prepared materials ZA-1 and ZA-4.

Types of Bacteria	Diameter of Inhibition Zone (mm)			
	Sample ZA-1		Sample ZA-4	
	D	W	D	W
*K. pneumonae*-1	7.5	NA	8.4	NA
*S. aureus*	NA	NA	7.5	8.7
*E. coli*	8.5	NA	7.7	7
*P. aeruginosa*	15	NA	16.5	NA

**Table 5 nanomaterials-12-02749-t005:** Comparison between the antimicrobial activity of the prepared materials ZA-1 and ZA-5.

Types of Bacteria	Diameter of Inhibition Zone (mm)			
	Sample ZA-1		Sample ZA-5	
	D	W	D	W
*K. pneumonae*-1	7.5	NA	12.5	NA
*S. aureus*	NA	NA	NA	NA
*E. coli*	8.5	NA	16.7	NA
*P. aeruginosa*	15	NA	13.7	NA

## Data Availability

Data supporting the reported results will be provided by the authors upon request.

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
