# Peer review of "Impact of Nanolayered Material and Nanohybrid Modifications on Their Potential Antibacterial Activity"

_nanomaterials, 2022, doi:10.3390/nano12162749_

Round 1

Reviewer 1 Report

The objective of the work is to study new modified layered double hydroxide materials with  effective antimicrobial activity against a wide range of gram-negative and gram-positive bacteria. Several modifications were used for developing the nanolayered structures of LDHs and organic species were intercalated inside the LDHs for controlling the interlayered spacing of the nanolayered structures, as well as carbon nanotubes were used as filler for the LDHs to increase the antibacterial performance. Five samples of Al/Zn nanolayered materials were prepared and modified and the materials were characterized through XRD,EDX, FTIR, DSC and TGA techniques. In the second part of the manuscript authors tested antimicrobial activity of nanocomposite material modified through the agar diffusion method. In particular, the ZA-3 sample AND ZA-5 (with stearic acid) showed important antimicrobial effects against E. coli, a common microorganism in the wastewaters and the ZA-4, with CNT, has a good effect on P. aeruginosa. The manuscript presents interesting results and can be published, but before publication I have some questions and suggestions. The language must be improved. I give some suggestions (some are in capital letters) and I raise questions:  

1. In the Abstract section, I think these phrases are not necessary: "The increase of bacterial resistance against multiple antibiotics is one of the main challenges for the scientific society. Therefore, new solutions were applied for developing materials with antimicrobial properties to prevent infectious diseases and inhibit the spread of bacteria especially that are commonly found in the environment and wastewater."  

2. Additionally, I have a suggestion: "Nanolayered structures and nanohybrids WERE prepared and modified by several techniques..."   In the Results section:

3. "X-ray diffraction (XRD) has BEEN used to identify the layered structure..."  

4. I did not understand these phrases: "The XRD diagram showed the characteristic peaks of the layered structure of the natural hydrotalcite for the sample ZA-1 agreeing with the standard file of JCPDS No. 37-629. Also, it matches with the standard file of JCPDS No. 48-1022 of the synthetic Zn–Al LDH as shown in Figure 1a." Is the diffractogram agreeing with two different materials? Or are these associated with ZA-1 and ZA-2? Please, explain this point in a better manner.  

5. In line 245 I suggest: "These EVENTS can be attributed to the decomposition of the.."  

6. Looking at the TG curve of ZA-1 we note that the onset of the first loss of mass occurs at about 120 oC. Is this correct?  

7. Authors say: "The decomposition of carbonate and cyanate anions was observed in the range of 161-310°C as shown in Figure 1d." How is it possible to associate these compositions with the two temperatures?   

8. Authors say: "Results shown in Figure 1 and presented in Table 1 indicated that ZA-1 has an inhibitory effect on all tested bacterial strains. such as Klebsiella pneumoniae, Staphylococcus aureus, E. coli and P. aeruginosa especially when dissolved in DMSO." However, in Table 1, we observe that there is no inhibitory effect on S. aureus. Please, correct it.  

9. I suggest in the caption of Fig. 3(c) adding the meaning of the yellow arrows.  

10. In page 299: "In our study, ultrasonic technique has BEEN used to modify and develop the size and the nanolayered structure..."  

11. In the captions of Fig. 3(a), 5(a) and 6(a) I suggest writing something like this: (a) X-ray diffraction (numbers represent d-spacing).   

12. Regarding the ZA-4 sample, is the CNT present between the two structures? Is it possible to understand its localization in the sample?  

13. Regarding the ZA-5 sample, authors write: "In addition, Figure 6a showed broad peak at 0.45 nm indicating presence of poly vinyl alcohol inside the nanolayered structure of ZA-5." Why does the existence of this peak indicate the presence of poly vinyl alcohol inside the nanolayered structure?  

14. Suggestion in p. 527/ 528: "The second type of hydroxyl groups, which IS affected by hydrocarbon chains BELONG to the nanolayered..."  

In conclusion, there are some questions to be answered and I also suggest a rigorous review in the English language. 

Author Response

Responses to reviewer comments and suggestions attached

Reviewer 2 Report

The manuscript describes the synthesis, characterization of modifications of nanomaterials and their potential antibacterial activities. It is an interesting topic which I consider of interest for readers of Nanomaterials.

The manuscript needs major revision before its accepted for publication.

Comments:

-          An illustration on Layered Double Hydroxide (LDH) will be helpful in the introduction

-          The aim of the study in introduction can be in a new paragraph

-        In Results section, Need a small paragraph to describe the synthesis.  Suggest to include a table that summarizes the different characteristics of ZA-1 to ZA-5.

-         Results are a bit confusing, suggest to explain on characterization results (XRD, EDX Spectra, thermal analyses) first in one section;  followed by antimicrobial activity in another section (table 1 – 4 can also be combined to a single table in this way for easier comparison for readers)

-          Certain results/data can be put to supplementary paper (eg. FITR, thermal analyses, characterization results)

-          Suggestions to write a few words for future studies in the discussion part.

-          Typo at line 529, p14. “Nanohybrid”

Author Response

Response to the reviewer comments:

Kindly find attached file 

Round 2

Reviewer 2 Report

All my comment have been addressed. 

Even though my suggestion to put the data of characterization (XRD, EDX Spectra, thermal analyses)  in one section;  followed by antimicrobial activity in other section hasn't been accepted by authors, it is okay that the manuscript keep the current form.